# Response to comment on 'AIRE-deficient patients harbor unique high-affinity disease-ameliorating autoantibodies'

**Christina Hertel[1], Dmytro Fishman[2], Anna Lorenc[3], Annamari Ranki[4], Kai Krohn[4], Pärt Peterson[5], Kai Kisand[5]\*, Adrian Hayday[3,6]\***

[1]ImmunoQure AG, Dusseldorf, Germany; [2]Institute of Computer Science, University of Tartu, Tartu, Estonia; [3]Peter Gorer Department of Immunobiology, King's College London, London, United Kingdom; [4]Department of Dermatology, Allergology and Venereology, Institute of Clinical Medicine, University of Helsinki, Helsinki, Finland; [5]Institute of Biomedicine and Translational Medicine, University of Tartu, Tartu, Estonia; [6]Francis Crick Institute, London, United Kingdom

**Abstract** In 2016, we reported four substantial observations of APECED/APS1 patients, who are deficient in AIRE, a major regulator of central T cell tolerance (Meyer et al., 2016). Two of those observations have been challenged. Specifically, 'private' autoantibody reactivities shared by only a few patients but collectively targeting >1000 autoantigens have been attributed to false positives (Landegren, 2019). While acknowledging this risk, our study-design included follow-up validation, permitting us to adopt statistical approaches to also limit false negatives. Importantly, many such private specificities have now been validated by multiple, independent means including the autoantibodies' molecular cloning and expression. Second, a significant correlation of antibody-mediated IFNα neutralization with an absence of disease in patients highly disposed to Type I diabetes has been challenged because of a claimed failure to replicate our findings (Landegren, 2019). However, flaws in design and implementation invalidate this challenge. Thus, our results present robust, insightful, independently validated depictions of APECED/APS1, that have spawned productive follow-up studies.
DOI: https://doi.org/10.7554/eLife.45826.001

**\*For correspondence:**
kai.kisand@ut.ee (KK);
adrian.hayday@kcl.ac.uk (AH)

## Introduction

In 2016, in a paper published in *Cell*, we made at least four substantial observations concerning APECED/APS1 syndrome patients who are defined by deficiency in the *AIRE* gene, a major regulator of central T cell tolerance: i) that such patients share autoantibodies to a small subset of proteins, including Type I IFNs, Interleukin-(IL)−17, and IL-22, that was previously reported (*Kisand et al., 2010*; *Meager et al., 2006*); ii) that, quite surprisingly, many such naturally-arising antibodies are conformation-specific and of extremely high affinity, potentially explaining their powerful neutralizing capacity; iii) that most APECED/APS1 patients additionally harbor 'private reactivates' collectively targeting very many autoantigens; and iv) that strong antibody-mediated neutralization of IFNα correlated significantly with an absence of Type I diabetes (T1D) in patients otherwise highly disposed to it (*Meyer et al., 2016*). These observations formed the basis for several substantial follow-up papers in peer-reviewed journals (*Rodero et al., 2017a*; *Frémond et al., 2017*; *Fishman et al., 2017*).

Our 2016 paper discussed the close alignment of our first substantial observation with contemporaneous work by Landegren and co-workers that likewise featured a protein microarray screen of patient *versus* control sera (*Landegren et al., 2016*). Our second substantial observation could not

be compared because Landegren and co-workers did not investigate the properties of the autoantibodies that were identified. Nonetheless, Landegren and other co-workers now dispute our remaining two claims (*Landegren et al., 2019*). While we welcome open, constructive discourse about science, we are disappointed by this dispute because we believe it reflects simple but important differences between our approaches that could have been easily resolved, had Landegren and co-workers approached us directly. Those important differences are explained below. Based on biological significance, they are considered in reverse order to their consideration in *Landegren et al. (2019)* (hereafter referred to as the comment).

## Results and conclusions

### No association between neutralizing autoantibodies to interferons and Type 1 diabetes in APECED/APS1

The comment disputes our observed correlation of strongly neutralizing IFNα autoantibodies with reduced incidence of T1D, claiming to have essentially repeated our experiment, but finding no difference in the IFN neutralization capacity of sera from APECED/APS1 patients with or without T1D.

In fact, the comparison that is described in the comment of IFN neutralization in two APECED/APS1 patient cohorts defined simply as with or without diabetes did not repeat our experiment. We did not claim that differential Type I IFN neutralization is the sole regulator of T1D incidence. Rather, we hypothesized that the significance of differential Type I IFN neutralization might relate to the differential disease development in patients uniformly displaying pathognomonic features of T1D. Thus, we compared IFNα neutralization in two patient sub-cohorts: one presenting with T1D and one not, but all of whom either harbored and/or had harbored disease-associated, anti-islet antibodies, for example anti-GAD65, anti-GAD67, that are widely-utilized clinical indicators of T1D-risk. Supporting our hypothesis, we observed a statistically significant correlation of low neutralization and T1D, consistent with several other experiments in which we established the capacity of APECED/APS1 autoantibodies to ameliorate immunopathology.

It is also unfortunate that an imperfect study design was employed in the comment. As presented to us, the comparison of different sera in the comment was made at a single [high] concentration (10%), which is inappropriate because it will most often saturate the assay, thereby failing to appropriately discriminate low *versus* high neutralizing activities. To illustrate this point, we re-examined IFNα neutralization, using a dose-dependent, cell-based assay that measures IFN-stimulated release of alkaline phosphatase (AP), as we previously described (*Meyer et al., 2016*), but mimicked the comment in examining only the effects of 10-fold diluted sera. This masked any significant differences between GADA-seropositive patients with or without T1D (*Figure 1A*). By contrast, when sera were serially diluted so as to imbue the assay with appropriate sensitivity (as described in the *Meyer et al., 2016*), the patients' broad dynamic range of IC50 values was revealed, with clear segregation of the patients with and without T1D (*Figure 1B and C*). Indeed, among 13 patients without T1D, the serum of only one ('N'; *Figure 1C*) showed low IFNα neutralization, comparable to that of all the patients with T1D.

The authors of the comment employed a phospho-STAT1 induction assay (*Gupta et al., 2016*). This is an inherently less sensitive assay, but nonetheless when we adopted it in another new experiment, we obtained the same pattern of results as we obtained with the AP assay. Namely, at high concentrations, the sera of patients with and without T1D showed comparable activities, but at lower, sub-saturation concentrations [50-fold dilutions], the cohort without T1D showed significantly greater capacity to limit IFNα activity (*Figure 1D*). Thus, because their measurements were insufficiently sensitive to discriminate low neutralizers from high neutralizers, we believe that the experimental design employed in the comment was not appropriate to compare IFN neutralization by the sera of patients with and without T1D: as such, the comment provides no experimental basis on which to dispute the fourth substantial observation of *Meyer et al. (2016)*.

Finally, the observations of *Meyer et al. (2016)* are germane to an important clinical issue. Specifically, the delayed onset and relatively rare incidence (~15%) of T1D in APECED/APS1 patients is puzzling given that: insulin is a prototypic AIRE-regulated tissue-specific autoantigen; there is

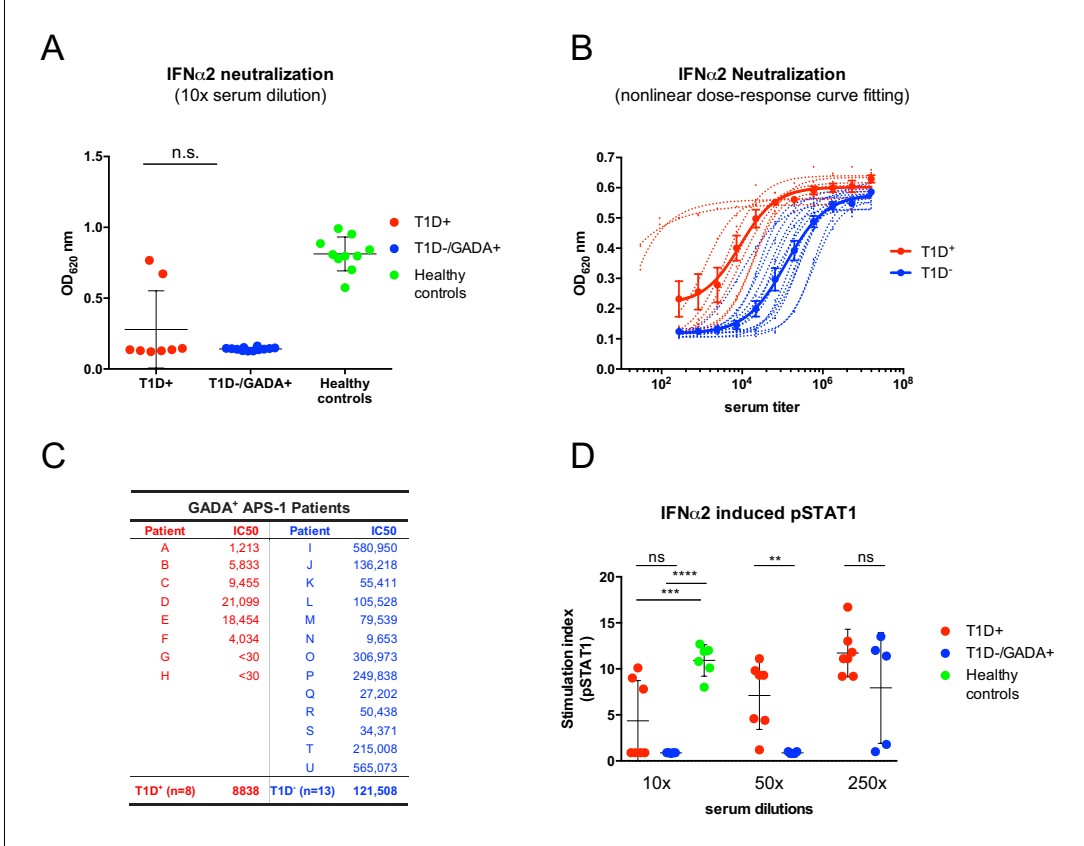

**Figure 1.** The comparison of two different strategies to measure IFNα neutralizing capacity of APECED/APS1 serum samples. In panel (**A**), the same reporter cell assay (HEK-Blue IFN-α/β cells from InvivoGen) has been applied as in **Meyer et al. (2016)** but at a single high serum concentration (ns: not significant). (**B**) Representative fitted dose-response curves that were used in **Meyer et al. (2016)** to calculate IC50 values for each serum sample. Individual curves are represented with dotted lines and those for grouped values in solid lines (mean ± SEM). (**C**) IC50 values (expressed as the dilution of serum sufficient to neutralize 50% of IFNα2 activity [12.5 U/ml]) that were calculated from individual and grouped curves shown in panel B. APECED/APS1 patients with Type 1 diabetes (T1D) are depicted in red and APECED/APS1 patients with GAD65 autoantibodies (GADA) but without T1D are in blue. (**D**) Neutralization of IFNα2 activity (10 000 U/ml) required to induce pSTAT1 was tested with different dilutions of sera from GAD seropositive patients with and without T1D. 2-way ANOVA was used to calculate P-values [ns – not significant, **p≤0.01, ***p≤0.001, ****p≤0.0001].
DOI: https://doi.org/10.7554/eLife.45826.002

The following source data is available for figure 1:

**Source data 1.**
DOI: https://doi.org/10.7554/eLife.45826.003

defective negative selection of β-cell antigen-specific T cells; pancreatic β-cells are notoriously vulnerable to autoimmune attack; and idiopathic T1D commonly occurs in children and adolescents (**Perheentupa, 2006**; **Anderson et al., 2002**; **Wolff et al., 2014**; **Sabater et al., 2005**). In this context, the observations of **Meyer et al. (2016)** suggest that IFN-neutralizing antibodies may delay T1D onset in APECED/APS1 patients and may prevent it completely in those with very high neutralizing titres. This is consistent with longitudinal assessment, albeit limited, reported by in Supplementary Figure 7 of **Meyer et al. (2016)**. When combined with increasing numbers of studies implicating Type I IFN as pathogenic in patients at genetic risk to develop T1D (**Ferreira et al., 2014**; **Kallionpää et al., 2014**; **Foulis et al., 1987**), our data compel us to disagree with the assertion made in the comment that there is insufficient evidence to "embark on in-depth investigations of targeting Type 1 IFNs for the treatment or prevention of Type 1 diabetes."

## No evidence for widespread autoantibody reactivity in APECED/APS1 patients

The comment disputes our observations that individual APECED/APS1 patients harbor small numbers of 'private' specificities shared by few other patients, but collectively comprising a very large number of proteins.

In fact, our observations conspicuously mirror a key clinical aspect of APECED/APS1, namely that each patient is highly individual in the type and range of symptoms; the rate and course of disease progression; and, to some extent, the time-of-onset. Hence, it makes biological sense that individual patients would harbor correspondingly diverse antibodies as causes and/or biomarkers of discrete clinical courses.

Nonetheless, the comment argues that the private specificities comprise stochastic, irreproducible signals reflecting the high risk of false positives inherent in the statistical methods that we employed to analyse our ProtoArray data.

Importantly, all statistical approaches need to reflect a study's goals. For example, clinical trials use one type of statistical method to minimize type one errors (false positives) that might misleadingly indicate drug efficacy, while employing different statistical methods to minimize type two errors (false negatives) that might obscure adverse event(s). The approach advocated in the comment (and in *Landegren et al., 2016*) parallels the former, scoring signals in patients by comparison to mean and standard deviation (SD) of controls, and then additionally adding a Fisher's exact test to exclude signals that did not confer statistical difference to the whole patient group. This is well-suited to defining how APECED/APS1 patients differ as a group from healthy controls.

By contrast, *Meyer et al. (2016)* sought to characterize the nature of auto-reactivities in APECED/APS1 patients, including private reactivities that might mirror individual clinical presentations. The Fisher's exact test filter would exclude 'private targets' as outliers or false positives because they are insufficiently frequent to significantly influence the distribution of reactivities across the whole group (see below). Anticipating this, and knowing that no existing standard data-analysis method can unequivocally discriminate private specificities from false positives, we compared the signal to mean and SD of controls without the additional filter (*Meyer et al., 2016*).

Although, this is a standard, widely-used approach, we do not dispute that it can be confounded by unwarranted assumptions about the behavior of the control cohorts, coupled with an imbalance in the numbers of controls (21) and patients (81) that we examined (*Meyer et al., 2016*). Hence, false positives can and will arise. Nonetheless, we consciously employed this approach because the validation of real reactivities *versus* false positives was to be made by a spectrum of additional, independent, serological, biochemical, and biological methods that were employed in *Meyer et al. (2016)* and subsequently in *Frémond et al. (2017)*. By contrast, in the comment (and in *Landegren et al., 2016*) the authors went little beyond the ProtoArray, necessitating their adoption of a more conservative statistical approach.

Examples of validation are as follows. First, several private anti-cytokine reactivities were validated by ELISA and by LIPS (Luciferase Immunoprecipitation – an unrelated assay platform using independent sources of target proteins displayed in native conformation), and have since been validated independently (*Meyer et al., 2016*; *Fishman et al., 2017*; *Sng et al., 2019*). Furthermore, we molecularly cloned and fully characterized such autoantibodies, for example anti-IL32γ (*Meyer et al., 2016*), anti-BAFF (unpublished data), and anti-IL20 (*Meyer et al., 2016*) detected in five, four, and two patients, respectively, but in none of the sampled controls.

Second, LIPS likewise validated many non-cytokine targets, including but not limited to 24 of 31 testis- and cancer-associated antigens so far tested (*Table 1*), commonly with good correlation with the ProtoArray signal intensities (*Fishman et al., 2017*). Those validated targets included twelve testis-specific and CT-antigens (PDILT, MAGE-B2, SPANXD, SPAG8, SPAG16, CT45A3, GAGE1, GAGE7B, MAGE-B1, MAGE-A3, MAGE-4 and MAGE-A10) (*Fishman et al., 2017*). This overtly contrasts with the comment and with *Landegren et al. (2016)* in which the ProtoArray analysis identified reactivity to only two CT-antigens (PDILT and MAGE-B2), providing experimental evidence that their statistical methods were too conservative to detect patients' private reactivities.

Third, *Fishman et al. (2017)* applied very stringent criteria to the data of *Meyer et al. (2016)*, including a further filtration of private reactivities into those shared by ≥3 patients. Still there were ~1000 reactivities: 490 shared by only three patients; 245 shared by 4 but not five patients; 111

**Table 1.** Testis- and cancer- associated non-cytokine targets screened by LIPS.

| Target | LIPS result |
| --- | --- |
| SPAG8 | pos |
| SPANXD | pos |
| TEX264 | pos |
| CT45A3 | pos |
| GAPDHS | pos |
| SPAG16 | pos |
| PDILT | pos |
| GAGE1 | pos |
| SPATA7 | pos |
| GAGE7 | pos |
| CAPNS1 | pos |
| KCNIP2 | pos |
| POMZP3 | pos |
| MAGEA4 | pos |
| RPL12 | pos |
| MKNK2 | pos |
| S100A7A | pos |
| MAGEA3 | pos |
| MAGEB1 | pos |
| MAGEB2 | pos |
| MAGEA10 | pos |
| LCN1 | pos |
| FGF12 | pos |
| HMGB1 | pos |
| TSPY2 | neg |
| MORN2 | neg |
| CRYGD | neg |
| GNG4 | neg |
| RSU1 | neg |
| PAGE1 | neg |
| PAGE2 | neg |

DOI: https://doi.org/10.7554/eLife.45826.004

shared by 5 but not six patients; 116 shared by >6 patients. These reactivities individually and collectively displayed five conspicuous traits: (1) correlations with clinical phenotypes, for example pernicious anemia or vitiligo; (2) more reactivities in patients with more complex clinical phenotypes; (3) a correlation of the average number of reactivities per patient with the severity of the AIRE gene mutation; (4) reactivities assessed longitudinally over relatively short time-frames correlated more closely than those sampled over longer time-frames (e.g. 10 years); and (5) reactivities mostly increased with duration of disease (*Fishman et al., 2017*).

Fourth, the reactivities described by *Meyer et al. (2016)* were conspicuously enriched in gene-products of two sub-classes: a) those expressed in lymphoid tissues and with no known connection to AIRE function, but which comprise some of the strongest reactivities (as agreed by *Landegren et al., 2016* and the comment); b) diverse tissue-restricted antigens (TRAs), which were strikingly enriched in those expressed by AIRE-expressing medullary thymic epithelial cells (*Fishman et al., 2017*). Consistent with this, male antigens were also targeted in females

(*Fishman et al., 2017*), whereas non-CT-antigen members of the MAGE family that are expressed in all tissues were not observed as targets (*Meyer et al., 2016*; *Fishman et al., 2017*).

False positives could not meet any of these four sets of criteria, let alone all of them. In sum, the potential for a signal to be a false positive does not establish that it is, particularly when its validity is attested to by multiple independent means. The comment ignores another pitfall of ProtoArrays, which is the under-estimation of reactivities to proteins that are not displayed well, as we and others have noted (*Kärner et al., 2016*; *Schnack et al., 2008*). Critically, ProtoArrays should serve as guides for subsequent experiments, as was the case for *Meyer et al. (2016)* and a number of later studies (*Rodero et al., 2017b*; *Frémond et al., 2017*; *Fishman et al., 2017*).

In this regard, we note that a co-author of the comment recently published a study (*Sng et al., 2019*) describing a loss of B cell tolerance in APECED patients that was associated with a broad spectrum of autoantigen reactivities, including several new non-cytoline specificities. This aligns with the depiction of APECED/APS1 patients provided by *Meyer et al. (2016)*.

Conceding, nonetheless, that we may have exaggerated some patient reactivities, we applied a more conservative statistical approach to *Meyer et al. (2016)*: namely we based z-scores on the mean of the controls and SD across all patients plus controls. SD will now be increased by positive reactivities in controls and/or patients, thereby reducing the risk of false positives. Interestingly, this approach identified reactivities overlapping 81% with our original study: again, these comprised broadly shared autoantigens and from ~30 to~100 private specificities that collectively composed a substantial fraction of the proteome. Moreover, when this same statistical approach was applied to an additional study in which we used an earlier version of the ProtoArray (v5.0) to interrogate sera from 23 patients examined by *Meyer et al. (2016)* but with eight different healthy controls, the overlap across the two independent studies (and platforms) was substantial and highly significant ($p < 1e\text{-}06$), far exceeding any overlap obtained from 100,000 random permutations of patients and controls.

We conclude that our published and ongoing studies (*Meyer et al., 2016*; *Fishman et al., 2017*) accurately depict the serological status of APECED/APS1 patients, viewed collectively and individually. While we acknowledge that the limited numbers of patients and appropriate controls make it difficult to reach a precise estimate of the numbers of private specificities, there is no basis for disputing the four central findings of *Meyer et al. (2016)*, consistent with which those findings have formed a basis for rigorous follow-up work by us and by others (*Rodero et al., 2017a*; *Frémond et al., 2017*; *Fishman et al., 2017*; *Sng et al., 2019*; *Rice et al., 2018*; *Llibre et al., 2018*; *Dhir et al., 2018*; *Rodero et al., 2017b*) that will inform our understanding of APECED/APS1 and of autoimmune diseases more generally.

## Materials and methods

### Key resources table

| Reagent type or resource | Designation | Source | Identifiers |
|---|---|---|---|
| Cell line | Human HEK293 cells - Type I IFNs reporter cells | InvioGen | cat # hkb-ifnab |
| Antibody | Alexa Fluor 647 conjugated anti-STAT1 (pY701), mouse IgG2a | BD Biosciences | cat # 562070 |
| Recombinant protein | recombinant human IFNα2a | Miltenyi Biotech | cat # 130-093-873 |

### Reporter cell assay

The IC50 values of IFNα neutralization of serum samples were tested with the help of HEK-BlueTM IFN-α/β reporter cells (InvivoGen) that express alkaline phosphatase (AP) under the inducible ISG54 promoter after ISGF binding to the IFN-stimulated response elements in the promoter. The cells were grown in DMEM supplemented with heat inactivated 10% FBS and 30 g/ml blasticidin (InvivoGen) and 100 g/ml Zeocin (InvivoGen). Cells were stimulated with IFNα2a (12.5 U/ml, Miltenyi Biotech) that was preincubated for 2 hr with serial dilutions of recombinant antibodies or one fixed

concentration (10%) of serum. QUANTI-Blue TM (InvivoGen) colorimetric enzyme assay was used to determine AP in the cell culture supernatants after 21 hr of incubation. OD was measured at 620 nm with Multiscan MCC/340 (Labsystems) ELISA reader and IC50 values were calculated from the dose-response curves using GraphPad Prism eight software.

## Phospho-STAT1 assay

Peripheral blood mononuclear cells (PBMC) from a healthy control were isolated with density gradient centrifugation and aliquoted by 500 000 cells to test tubes containing IFN-α2a (10 000 U/ml) pre-incubated with serum dilutions for 2 hr. Tubes with or without IFN alone served as positive and negative controls. After 15 min of stimulation of PBMCs at 37°C, the cells were fixed immediately with Cytofix buffer, permeabilized with Perm Buffer III and stained with PE-conjugated antibody to phospho-STAT1 (Y701; all from BD Biosciences). Data were acquired with LSRFortessa (BD Biosciences) and analyzed with FCS Express (De Novo Software).

# Additional information

## Competing interests

Christina Hertel: CH is an employee in ImmunoQure AG, which contributed research funding to the study by Meyer et al., 2016. Annamari Ranki: AR is an equity holder in ImmunoQure AG. Kai Krohn: KKr is an equity holder in ImmunoQure AG. Pärt Peterson: PP is an equity holder in ImmunoQure AG. Kai Kisand: KKi is an equity holder in ImmunoQure AG. Adrian Hayday: AH is an equity holder in ImmunoQure AG. The other authors declare that no competing interests exist.

## Funding

| Funder | Grant reference number | Author |
| --- | --- | --- |
| Wellcome Trust | 106292/Z/14/Z | Adrian Hayday |
| Estonian Research Competency Council | IUT2-2 | Pärt Peterson Kai Kisand |

The funders had no role in study design, data collection and interpretation, or the decision to submit the work for publication.

## Author contributions

Christina Hertel, Investigation, Writing—original draft, Writing—review and editing; Dmytro Fishman, Anna Lorenc, Writing—review and editing; formal analysis; statistical evaluations; Annamari Ranki, Kai Krohn, Data curation, Formal analysis; Pärt Peterson, Writing—original draft, Writing—review and editing; Kai Kisand, Investigation, Writing—original draft, Project administration, Writing—review and editing; Adrian Hayday, Conceptualization, Writing—original draft, Writing—review and editing

## Author ORCIDs

Dmytro Fishman (iD) https://orcid.org/0000-0002-4644-8893
Pärt Peterson (iD) https://orcid.org/0000-0001-6755-791X
Kai Kisand (iD) https://orcid.org/0000-0002-5426-4648

## Ethics

Human subjects: Use of human material was approved by local ethics committees (Finland: HUS Medical ERB, 8/13/03/01/2009. Slovenia: National Medical Ethics Committee number 22/09/09 and 28/02/13. Italy: Ethics Committee Prot. PG/2015/20440. Norway: Research Ethics Committee of Western Norway, health registry number 047.96, bio-bank number 2013-1504, project number 2012/1850. Estonia: Research Ethics Committee of the University of Tartu, 235/M-23). All individuals included signed informed consent.

**Decision letter and Author response**

Decision letter https://doi.org/10.7554/eLife.45826.010
Author response https://doi.org/10.7554/eLife.45826.011

## Additional files

### Supplementary files

• Transparent reporting form
DOI: https://doi.org/10.7554/eLife.45826.005

### Data availability

All data generated or analysed in this study are included in the manuscript and supporting files. ProtoArray data have been deposited at Array Express (E-MTAB-5369).

The following previously published dataset was used:

| Author(s) | Year | Dataset title | Dataset URL | Database and Identifier |
|---|---|---|---|---|
| Dmytro Fishman, Pärt Peterson | 2017 | Protein microarray (Protoarray, Invitrogen) screening with APECED patient, healthy relative and healthy control serum samples | https://www.ebi.ac.uk/arrayexpress/experiments/E-MTAB-5369/ | ArrayExpress, E-MTAB-5369 |

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
