## [Decision Letter]

Thank you for submitting your article "Response to Comment on 'AIRE-deficient patients harbor unique high-affinity disease-ameliorating autoantibodies'" to *eLife* for consideration as Scientific Correspondence. Your article, and the original comment by Landegren and colleagues, have been reviewed by two peer reviewers (who have opted to remain anonymous), and the evaluation has been overseen by a Deputy Editor (Detlef Weigel) and the *eLife* Features Editor (Peter Rodgers).

I am pleased to be able to tell you that we have decided to accept for publication both the original comment and your response to it. The substantive comments in this decision letter will also be published as part of your article (and likewise for the comment from Landegren): you will be able to check these comments when you check the proof of your article.

You will need to make the following changes to your article:

[... ]

Note: The editor handling this manuscript asked the reviewers to answer a number of questions: these questions are shown below in italics; the comments from the reviewers are shown in Roman.

Reviewer #1:

Is the Formal Response substantial enough to merit formal publication? Do you have any substantive concerns about the Formal Response? Do you think that the authors of the original publication should be asked to publish a correction to their work?

The Formal Response addresses the main challenges by Landegren et al. The response correctly points out that there were four major findings of the Cell 2016 paper and that Landegren challenges two of these. I acknowledge that the Cell 2016 paper was a landmark paper of recent years and that the cytokine antibody findings that were validated with LIPS assays in that paper were very impressive.

The formal response by Hertel et al is appropriate in its criticism of one point of contention - the issue of type 1 IFN neutralizing autoantibodies and diabetes development. As mentioned above and in the response, Landegren have reached saturation in their type 1 IFN neutralizing autoantibody assay and, therefore, cannot make their conclusion that there are no differences in type 1 IFN neutralizing autoantibodies between the APECED patients with and without diabetes. The formal response and new data convincingly demonstrate the original claim by Meyer. One point to note, however, is that the majority of GAD antibody positive individuals (healthy or diseased) do not develop Type 1 diabetes (unless they have other Type 1 diabetes autoantibodies) and it is highly unlikely that many of these have high titers of type 1 IFN neutralizing autoantibodies - be careful about over-generalizing the findings from the APECED patients.

The second point of contention is less well defended. Indeed, one expects that many readers of the Cell paper would see the flaws in using Z scores with 21 control samples. This simply cannot be defended. The formal response argues that if they calculated Z scores using all data (patients and controls) they have similar findings, but still there are too few to be conclusive and the imbalance of controls and patients is not overcome by this analysis. The formal response correctly states and appropriately points out that the alternative approach to a threshold will miss private specificities and that they purposefully chose their analysis method to be able to address their question of how broad is the spectrum of autoreactivity in APECED patients. However, they would have needed to include hundreds (at least) of controls to do this.

The second problem is that the disease is rare and, therefore, one must consider private specificities (those seen in occasional patients) as suggestive, but not conclusive. The formal response uses the subsequent paper of Fishman as evidence to support their claims for many of the antigens of their screen. Unfortunately, on their second point of evidence, the Fishman paper shows correlations between the array and the LIPS assay Z scores (note that the LIPS Z scores are actually not very high), which is not entirely convincing as a validation of these antigens and there were only 20 controls. Thus, the formal response is subject to the same flaws - too few controls for Z scores in what they use as evidence to their original claim. Also, I have concerns about Table 1 in the formal response as I don't find the data it is derived from sufficiently conclusive. The formal response argues that they have other sets of evidence that support their claim of broad reactivity, namely their impressive validation of the cytokine autoantibodies in the original paper (and by Landegren), the tissue enrichment of their targets and the association of specificities to disease phenotype, and the more stringent filtering of data in the Fishman paper. These arguments are reasonable.

I do not believe that the authors of the original work should publish a correction, but I do believe that in their response they should have recognized the flaw in defining positivity with Z scores derived from 21 controls and that they should have stated even more clearly that their method of analysis is biased towards identifying false-positives. I believe that it would be fair and objective to say validation was performed for many of the cytokine targets in the original paper and only partially done for some of the other specificities in the Fishman paper. I also suggest that they should have been a little less aggressive in their response.

Specific questions for the original paper (Meyer et al. 2016 Cell 166:582–595) are:

Was the initial screen indeed sufficient to generate a list of candidates that were more likely than expected by chance to show an effect in the experimental follow-up studies?

The initial screen successfully identified and validated a number of cytokines as autoantigen targets. Thus, it was able to show that the screen was sufficient to give rise to a very important and repeatedly validated finding. However, it was inadequate to make the claim that thousands of antigens can be targeted in these patients. The ProtoArray is just not ideal for such a conclusion without testing all the antigens in a different system. It is also way too expensive to be used with hundreds (a minimum) of controls necessary to appropriately define Z scores, and APECED patients are simply too rare to have a sufficiently large cohort to be conclusive about low frequency positive targets. The follow-up studies that did test some of the antigens by LIPS don't do a lot to dampen these limitations since they also don't have many controls and patients.

Are the experimental follow-up studies believable?

The experimental follow-up studies on the cytokine antibodies (the major findings) are entirely credible with validation by ELISA and LIPS, cloning of IgG from memory B cells, neutralizing antibody assays, and subsequent validation by others. The follow-up studies for other antigens reported by Fishman are much less credible.

In rebuttal, the authors of the Cell paper claim that a sampling of the thousands of private specificities were validated. There was no evidence for this in the published study, and unconvincing preliminary evidence here.

Reviewer #2:

Is the Formal Response substantial enough to merit formal publication? Do you have any substantive concerns about the Formal Response? Do you think that the authors of the original publication should be asked to publish a correction to their work?

i) Is the Formal Response substantial enough to merit formal publication?

Yes.

ii) Do you have any substantive concerns about the Formal Response?

No.

iii) Do you think that the authors of the original publication need to publish a correction to their work?

Yes.

Please note that Reviewer #2 made the following comments about your response in their report about the submission from Landegren et al.

- Do you have any substantive concerns about the elements of the article [by Landegren et al.] that challenges the findings of the original publication?

No, the analysis performed by Landegren provide compelling evidence that the original publication in Cell employed a flawed normalization of the analysis of antibody binding to protein microarrays. As a result, the main conclusion in the Cell paper about the much wider breakdown of tolerance in AIRE deficiency, with antibodies recognizing private specificities unique to one or two patients, is erroneous.

In rebuttal, the authors of the Cell paper claim that a sampling of the thousands of private specificities were validated. There was no evidence for this in the published study, and unconvincing preliminary evidence here.

- Are the other elements of challenge [by Landegren et al.] significant enough to merit publication in eLife?

Yes. The second main novel conclusion of the Cell paper was that neutralizing antibodies to interferon were absent from the sera of AIRE deficient patients with Type 1 diabetes compared to those without. Here, studying a larger cohort, Landegren et al find neutralizing antibodies in AIRE deficient patients with or without Type 1 Diabetes. It is possible this reflects differences in assay methodology, but both methods appear equally valid and unlikely to explain the different results.

In rebuttal, the original authors point out that the data in the Cell paper were from serum titrations, and that if they repeat the experiments with serum diluted 1/10 they also now find neutralizing antibodies in both groups. In the Cell paper, the mean IC50 of neutralizing antibodies in patients without Type 1 diabetes was on the order of 1/100,000, at which point there was a difference in titre. However in the data in the rebuttal, the difference in inhibitory activity is only apparent at one dilution (1/50) and not at 1/250 or 1/10. Since the rebuttal now also shows neutralizing antibodies are present in Type 1 diabetes patients at relatively small differences in titre, and since serum is undiluted in vivo, this appears to warrant a revision to the original conclusions in Cell that "those with T1D showed only low or negligible neutralization."

It is nevertheless difficult to extrapolate serum neutralization from in vitro to in vivo, so it would be important for Landegren et al to test interferon alpha neutralization at different serum dilutions to determine if there is a difference in titre in their cohort.

---

## [Author Response]

[This document contains the response from Hertel et al. to specific points in the decision letter sent on 9 April 2019; the full decision letter is available at the following URL: https://elifesciences.org/articles/45826#SA1]

We repeat the reviewers’ points here in italic, and include our replies point by point in Roman.

Reviewer #1:

The formal response by Hertel et al is appropriate in its criticism of one point of contention - the issue of type 1 IFN neutralizing autoantibodies and diabetes development. As mentioned above and in the response, Landegren have reached saturation in their type 1 IFN neutralizing autoantibody assay and, therefore, cannot make their conclusion that there are no differences in type 1 IFN neutralizing autoantibodies between the APECED patients with and without diabetes. The formal response and new data convincingly demonstrate the original claim by Meyer. One point to note, however, is that the majority of GAD antibody positive individuals (healthy or diseased) do not develop type 1 diabetes (unless they have other type 1 diabetes autoantibodies) and it is highly unlikely that many of these have high titers of type 1 IFN neutralizing autoantibodies - be careful about over-generalizing the findings from the APECED patients.

REPLY: We agree with the reviewer, and make clear in our response that neither we nor anyone has suggested that natural Type I IFN neutralization is a widespread means of naturally limiting disease. This notwithstanding, the comparison by Meyer et al. of GAD-reactive T1D^+^ and T1D^-^ patients, together with the animal studies conducted, provides a powerful illustration of the potential of autoantibodies to be disease-ameliorating, and a justification for examining such possibilities in other settings.

Unfortunately, on their second point of evidence, the Fishman paper shows correlations between the array and the LIPS assay Z scores (note that the LIPS Z scores are actually not very high), which is not entirely convincing as a validation of these antigens and there were only 20 controls. Thus, the formal response is subject to the same flaws - too few controls for Z scores in what they use as evidence to their original claim. Also, I have concerns about Table 1 in the formal response as I don't find the data it is derived from sufficiently conclusive.

REPLY: We absolutely do not agree that relatively low z scores for LIPS assays are of concern. For example, CYP21, which is a very well defined, broadly accepted autoantigen also has a low z score in LIPS. Importantly, these scores are significant. We do not show raw data in Table 2, because the relevant information is provided in reference 4. However, we are happy to provide the raw data, should it be requested. Importantly, the many autoantigen specificities that Meyer et al described and that are further studied by Fishman et al., 2017 have many key properties (e.g. correlation with specific clinical disease course) that are laid out in our response and that could not be expected for false positives.

I do not believe that the authors of the original work should publish a correction, but I do believe that in their response they should have recognized the flaw in defining positivity with Z scores derived from 21 controls and that they should have stated even more clearly that their method of analysis is biased towards identifying false-positives. I believe that it would be fair and objective to say validation was performed for many of the cytokine targets in the original paper and only partially done for some of the other specificities in the Fishman paper.

REPLY: We extensively discussed the issue of false positives in our paper, and have devoted additional attention to this in our response, acknowledging that our choice of statistical method risked false positives as a cost of limiting false negatives. By contrast, the original report by Landegren et al., 2016 is very vulnerable to false negatives, leading them to claim that private reactivities do not exist, despite the convincing demonstration of several of them by us and, more recently, by co-authors of Landegren et al (Sng et al., 2019).

I also suggest that they should have been a little less aggressive in their response.

REPLY: We believe that we have made our case firmly but not aggressively.

However, it was inadequate to make the claim that thousands of antigens can be targeted in these patients. The ProtoArray is just not ideal for such a conclusion without testing all the antigens in a different system. It is also way too expensive to be used with hundreds (a minimum) of controls necessary to appropriately define Z scores, and APECED patients are simply too rare to have a sufficiently large cohort to be conclusive about low frequency positive targets. The follow-up studies that did test some of the antigens by LIPS don't do a lot to dampen these limitations since they also don't have many controls and patients.

REPLY: This critique is intuitive rather than evidence-based. ProtoArrays have proved extremely important, hypothesis-generating guides to the status of sera and other biological samples, provided that those hypotheses are followed up with downstream analyses, as were conducted for cytokine reactivities by Meyer et al^1^ and for additional reactivities by Fishman et al^4^. When that is the case, it is fair to draw the community’s attention to conclusions that may be drawn from them. Nonetheless, we have never stated that we have independently validated all the proto-array “hits”, just as we cannot exclude the existence of additional false negatives that lead to an under-estimate of serum reactivity.

In rebuttal, the authors of the Cell paper claim that a sampling of the thousands of private specificities were validated. There was no evidence for this in the published study, and unconvincing preliminary evidence here.

REPLY: We repeat our response to the previous point.

Reviewer #2:

The analysis performed by Landegren provide compelling evidence that the original publication in Cell employed a flawed normalization of the analysis of antibody binding to protein microarrays. As a result, the main conclusion in the Cell paper about the much wider breakdown of tolerance in AIRE deficiency, with antibodies recognizing private specificities unique to one or two patients, is erroneous.

REPLY: First, a thorough reading of Meyer et al makes clear that the issue of private reactivities was not the major conclusion but one of four. Beyond this point, both the biologists and the statisticians on our team completely disagree with this Reviewer’s assessment. Just because the statistical approach knowingly adopted makes our study vulnerable to false positives does not mean that the identified private reactivities are false positives. This point is most convincingly made by our validation of several of the reactivties, to the point of cloning the autoantibodies responsible, and by our demonstration that the properties of many such specificities correlate with key features of disease progression and of AIRE-deficiency. For example, the autoreactivity of female patients toward male antigens, and the response to tissue-restricted MAGE proteins but not to ubiquitously expressed ones with similar physicochemical properties! Such properties confound the reviewers convictions over false positives. Moreover, a recent paper (Sng et al., 2019)authored by a prominent co-author of Landegren et al reports a wide-ranging breakage of tolerance in the B cell compartment, consistent with which they describe additional autoantigen specificities.

We should also note that while Landegren and the reviewers are able to criticize our statistical approach (as do we) for its vulnerability to false positives, none of them offers a constructive suggestion of a method to limit false positives while permitting the detection of private reactivities: the simple fact is that there is no silver bullet method.

In rebuttal, the authors of the Cell paper claim that a sampling of the thousands of private specificities were validated. There was no evidence for this in the published study, and unconvincing preliminary evidence here.

REPLY: As we responded above to Reviewer 1, this critique is not evidence-based. ProtoArrays have proved extremely important, hypothesis-generating guides to the status of sera and other biological samples, provided that those hypotheses are followed up with downstream analyses, as were conducted for cytokine reactivities by Meyer et al^1^ and for additional reactivities by Fishman et al^4^. When that is the case, it is fair to draw the community’s attention to conclusions that may be drawn from them. Nonetheless, we have never stated that we have independently validated all the proto-array “hits”, just as we cannot exclude the existence of additional false negatives that lead to an under-estimate of serum reactivity.

In rebuttal, the original authors point out that the data in the Cell paper were from serum titrations, and that if they repeat the experiments with serum diluted 1/10 they also now find neutralizing antibodies in both groups. In the Cell paper, the mean IC50 of neutralizing antibodies in patients without type 1 diabetes was on the order of 1/100,000, at which point there was a difference in titre. However, in the data in the rebuttal, the difference in inhibitory activity is only apparent at one dilution (1/50) and not at 1/250 or 1/10. Since the rebuttal now also shows neutralizing antibodies are present in Type 1 diabetes patients at relatively small differences in titre, and since serum is undiluted in vivo, this appears to warrant a revision to the original conclusions in Cell that "those with T1D showed only low or negligible neutralization.”

REPLY: Our response now provides expanded data-sets that use independent assays to provide unequivocal support for our claims. It is of course the case that the value of serum dilution required to see neutralizing activity will be different depending on the assay used, and will be higher for a less sensitive assay such as the STAT1 phosphorylation employed by Landegren et al. Indeed, because of its relative insensitivity, the pSTAT1 assay is not suitable for the generation of full dose-response curves that are necessary for precise estimation of neutralizing capacity. Nonetheless, whether we employed our highly sensitive assay for IFN-dependent alkaline phosphatase production or the pSTAT1 assay, we could demonstrate a clear and significant capacity to segregate the patients with T1D from those that did not. Hence, our conclusions are fully supported. Finally, we are extremely surprised by the Reviewer’s seeming assertion that IC50 may be irrelevant given that serum is undiluted in vivo. The key site of IFN activity may be the tissues, and it is completely unclear to what degree serum autoantibodies reach that site in “serum concentrations”.

In conclusion, Meyer et al^1^ draw the community’s attention to four major aspects of APECED biology. Two of these were unchallenged by Landegren et al., and the challenges made to the other two are insufficient to undermine them and the conclusions that we draw based on them. Those conclusions have provoked numerous follow up studies of benefit to biological understanding and to the elucidation of human pathophysiology.